# Cortical Function in Acute Severe Traumatic Brain Injury and at Recovery: A Longitudinal fMRI Case Study

**DOI:** 10.3390/brainsci10090604

**Published:** 2020-09-03

**Authors:** Karnig Kazazian, Loretta Norton, Teneille E. Gofton, Derek Debicki, Adrian M. Owen

**Affiliations:** 1Brain and Mind Institute, Western University, London, ON N6A3K7, Canada; uwocerc@uwo.ca; 2Graduate Program in Neuroscience, Western University, London, ON N6A3K7, Canada; 3Department of Psychology, King’s University College at Western University, London, ON N6A3K7, Canada; lnorton5@uwo.ca; 4Department of Clinical Neurological Sciences, Western University, London, ON N6A3K7, Canada; teneille.gofton@lhsc.on.ca (T.E.G.); derek.debicki@lhsc.on.ca (D.D.); 5Department of Physiology and Pharmacology and Department of Psychology, Western University, London, ON N6A3K7, Canada

**Keywords:** coma, consciousness, awareness, disorders of consciousness, traumatic brain injury

## Abstract

Differences in the functional integrity of the brain from acute severe brain injury to subsequent recovery of consciousness have not been well documented. Functional magnetic resonance imaging (fMRI) may elucidate this issue as it allows for the objective measurement of brain function both at rest and in response to stimuli. Here, we report the cortical function of a patient with a severe traumatic brain injury (TBI) in a critically ill state and at subsequent functional recovery 9-months post injury. A series of fMRI paradigms were employed to assess sound and speech perception, command following, and resting state connectivity. The patient retained sound perception and speech perception acutely, as indexed by his fMRI responses. Command following was absent acutely, but was present at recovery. Increases in functional connectivity across multiple resting state networks were observed at recovery. We demonstrate the clinical utility of fMRI in assessing cortical function in a patient with severe TBI. We suggest that hallmarks of the recovery of consciousness are associated with neural activity to higher-order cognitive tasks and increased resting state connectivity.

## 1. Introduction

Assessing cortical activity in critically ill brain injured patients following severe traumatic brain injury (TBI) is a complex clinical undertaking [1]. Bedside behavioural examinations are used to evaluate the functional integrity of the brain and by extension, the level of consciousness a patient retains in a critical care environment [2,3]. While considered to be the gold standard for evaluating consciousness, bedside clinical assessments may be confounded by the behavioural fluctuations of a patient, mechanical intubation, facial and ocular injuries, and poor inter-rater reliability [4,5]. A clear understanding of a patient’s cortical function in the intensive care unit is imperative as it informs critical decisions regarding the continuation of care or the withdrawal of life-sustaining therapy [6]. For these reasons, there is a pressing need for objective and reliable measures of cortical activity in critically-ill patients who have sustained a severe brain injury.

Functional magnetic resonance imaging (fMRI) may complement bedside behavioural examinations by providing objective measures of a patient’s neural activity in response to stimuli and at rest. A growing body of research has demonstrated the clinical utility of fMRI for assessing the cortical activity of patients in an intensive care setting [7,8,9,10,11,12,13]. Functional neuroimaging has also been used to evaluate cortical function in survivors of severe TBI who made a meaningful neurological recovery, providing insight into long-term cortical function after injury [14,15,16]. Evaluating which domains of brain activity change, and which remain similar, may provide valuable insight into the functional integrity of the brain as it moves from a behaviourally unresponsive to a conscious state.

Here, we present a case study of a patient who sustained a severe TBI resulting in acute intensive care treatment. The patient underwent functional neuroimaging to examine brain activity while in the ICU and again after recovery of consciousness 9-months post injury. We hypothesized that some aspects of basic perceptual cognition might be preserved in acute severe brain injury and detectable with fMRI. Moreover, we predicted increased activity in response to higher-order cognitive tasks and increased functional connectivity at recovery, reflecting the observed changes in behaviour.

## 2. Materials and Methods

### 2.1. Patient Information

The patient was a 34-year-old male involved in a high-impact motor vehicle collision. The patient was found unresponsive, but not submerged, in a body of water after being projected from his vehicle upon impacting a tree. He was admitted to hospital unconscious with polytraumatic injuries. Initial brain imaging (CT and a clinical anatomic MRI) revealed foci of intraparenchymal hemorrhage (the right midbrain and the left frontal lobe), intraventricular hemorrhage, and subarachnoid hemorrhage. Susceptibility weighted imaging also confirmed diffuse axonal injury (DAI) involving the cerebral hemispheres (Figure 1B), the corpus callosum, and the midbrain on the right side (Figure 1C). Other systemic injuries included pulmonary contusions; a moderate hemothorax; a diaphragmatic injury; hepatic injuries; a left renal laceration; and fractures to multiple ribs, the left clavicle, left L2 transverse process, and a non-displaced oblique fracture through the anterior arch of C1. Upon arrival to hospital, the patient was unconscious with a Glasgow Coma Scale (GCS) rating of 5 (eye = 1, verbal = 1T (intubated), motor = 3). Two days after hospital admission, the patient opened his eyes to auditory stimuli but did not fixate or track and was unable to obey commands. The patient’s pupils were equal, round, and reactive bilaterally (2 mm), with corneal reflexes present. The patient was not withdrawing to central pain, but sporadic spontaneous movement was noted.

The first fMRI scan in the acute phase of illness took place 26 days post-injury while the patient was still being treated in the intensive care unit. The patient was unable to undergo a research scan earlier than this due to his raised intracranial pressure and inability to lie flat in the scanner. At the time of the first scan, the patient had a GCS score of 8 (eye = 4, verbal = 1T, motor = 3). The patient’s eyes opened sporadically but he showed no evidence of fixation or tracking. He was unable to obey any commands, had spasticity in his four extremities, and was unable to localize to auditory stimuli. The patient remained intubated and mechanically ventilated at the time of imaging and was not receiving sedation. At day 37 post-injury, 11 days after the first fMRI, he regained some behavioural awareness with confusion and could localize to painful stimuli in all four limbs. Following gradual improvements in consciousness, the patient was transferred to an acquired brain injury rehabilitation inpatient program 73 days post-injury. The second fMRI scan took place 9 months post-injury and 2 months after discharge from rehabilitation. At the time of the second imaging session, the patient’s functional outcome was scored at 4 on the Glasgow Outcome Scale Extended (GOSE). The examination indicated that he had an overall moderate disability, with sections on the GOSE scored for lower severe disability (needing the help of someone around the home most of the time), upper severe disability (not able to shop without assistance and not able to travel locally without assistance), lower moderate disability (currently unable to work), and upper moderate disability (frequent disruption or strain on family due to psychological problems, participating much less than half as often in social and leisure activities). This assessment was obtained at a follow up appointment immediately before the fMRI scan by a certified neurocritical care clinician. A schematic timeline of the patient from acute care admission to the second scan is detailed in Figure 2. Full approval for the research study and procedures was granted by Western University’s Health Sciences Research Ethics Board (project identification code #101777). Stimuli in the auditory and command following tasks were delivered through MRI compatible sound-proof headphones.

### 2.2. fMRI Paradigms

#### 2.2.1. Auditory Perception

A passive, hierarchical, auditory perception paradigm was employed to assess sound and speech perception. The auditory perception task utilized an interleaved block design with four conditions; silence, signal correlated noise (SCN), meaningful speech, and “meaningless speech”, which is comprised of pseudowords that matched the words used in the meaningful speech condition in terms of syllable structure and transition frequency. Comparisons between variants of the meaningful speech and pseudowords conditions have been used previously to investigate language comprehension in group studies of healthy controls [17], but the contrast lacks sufficient power to yield reliable results in single-case studies. For that reason, these two tasks were “collapsed” for the purposes of this study and treated as a single “speech” condition. Each condition was thirty seconds in length and was repeated 5 times in a random order for a total of 10 min (4 conditions × 30 s × 5 repetitions).

#### 2.2.2. Command Following

A command following task requiring spatial navigation was employed to look for any evidence of covert awareness, as described elsewhere [18,19,20]. The participant was told to “imagine moving through your home and visualize the objects that you see” and to stop when hearing “now just relax”. An interleaved block design was used that alternated between 30 s of the command following tasks and 30 s of rest, totaling five and a half minutes in length for each paradigm (6 rest blocks, 5 imagery blocks).

#### 2.2.3. Resting State Connectivity

A resting-state fMRI scan was acquired to image the brain’s intrinsic functional connectivity at rest in the absence of any external stimuli. The resting state scan was five and a half minutes in length.

### 2.3. Neuroimaging Parameters

Imaging data was acquired on a 1.5T General Electric (Fairfield, CT) MRI machine located at London Health Sciences Centre (London, Canada). A whole-brain anatomical 3D-SPGR T1-weighted image was acquired with the following parameters: TR = 10.2 ms, TE = 4 ms, matrix size = 256 × 256, voxel size = 1.02 × 1.02 × 1.40 mm non-isotropic, flip angle = 10°. Functional imaging was acquired using the following parameters: TR = 2500 ms, TE = 40 ms, FA = 90 degrees, slice thickness = 5 mm, voxel size = 3.75 × 3.75 × 5mm non-isotropic and a T2*—weighted single-shot echo-planar imaging sequence to obtain 240 volumes of 30 slices in the auditory paradigm, and 132 volumes of 30 slices in the command following and resting state scans.

### 2.4. Analysis

#### 2.4.1. Pre-processing

SPM8 (Statistical Parametric Mapping: http://www.fil.ion.ucl/spm/software/spm8) was used to pre-process the imaging data. Functional images were spatially realigned for motion correction, co-registered to T1 structural images, and segmented and normalized to the SPM echo-planar imaging template. Functional images were spatially smoothed using an 8 mm full-width half-maximum Gaussian kernel.

#### 2.4.2. Tasked-Based Paradigm Analysis

In both the auditory perception and command following paradigms, a single-subject fixed-effect analysis was used to analyze the patient data in SPM8 (Statistical Parametric Mapping: http://www.fil.ion.ucl/spm/software/spm8). A general linear model (GLM) was created with regressors for each condition in each paradigm. Six individual motion parameters were included as covariates to account for variations due to movement. Based on the GLM, regressors for the auditory paradigm that corresponded to the presentation of silence, signal correlated noise, and speech conditions were created by convolving boxcar functions with the canonical hemodynamic response function. Regressors for the command following the paradigm were modelled to the canonical hemodynamic response function belonging to the spatial navigation or the rest condition. Analysis parameters were identical when assessing brain activity in the acute state of brain injury and at recovery. To assess for differences in activity between the two imaging sessions, the hemodynamic response from the recovery scan was contrasted against the acute scan for each paradigm (Recovery > Acute).

#### 2.4.3. Auditory Perception

Consistent with previously detailed methods [21], sound perception was assessed by comparing the auditory stimuli conditions (SCN, and two speech conditions) to the silent baseline to identify brain areas that were responsible for processing the acoustic properties of both speech and non-speech sounds. Changes associated with speech-specific perception were investigated by comparing the two speech conditions to the SCN condition. Results were thresholded at *p* < 0.05, family-wise error (FWE) corrected for multiple comparisons.

#### 2.4.4. Command Following

A region-of-interest (ROI) based approach was used to assess cortical responses in the spatial navigation task using a within-group analysis of previously published healthy control results for the same task [20]. The patient’s brain activity was analyzed at a peak voxel-wise threshold of *p* < 0.001, uncorrected. The ROIs were created using the SPM compatible MarsBaR software (http://marsbar.sourceforge.net/).

#### 2.4.5. Resting State Analysis

An independent component analysis was used to decompose the resting state blood-oxygen-level-dependent (BOLD) signal into 20 statistically independent spatial and temporal components with the GIFT software package (http://icatb.sourceforge.net). Components were then spatially correlated to 10 resting state network templates derived from the BrainMap database and included medial visual, occipital pole, lateral visual, default mode, cerebellar, sensorimotor, auditory, executive control, and the right and left frontoparietal networks [22]. In the patient data, the component with the greatest correlation to the spatial template was selected as the resting state network. A paired-samples *t*-test was used to determine if the mean correlation values between the acute and recovery scan across the 10 networks were statistically different.

## 3. Results

### 3.1. Auditory Perception

Acutely, the patient showed robust neural activity in response to both the sound and speech conditions (Figure 3A). In the sound perception contrast, activity was observed bilaterally in the temporal cortex, with peaks in both the left and right primary auditory cortices and superior temporal gyri (Table A1 in Appendix A). The strength of this activity was within the range of neural responses observed in healthy controls at the single-subject level in a prior study [13] (Figure 3C). In the speech perception contrast, the patient showed robust bilateral activity in the temporal cortex, with peaks in both the left and right superior temporal gyrus, left and right primary auditory cortex, and the left middle temporal gyrus (Table A2 in Appendix A). The extent of fMRI activity was once again comparable to that of healthy participants (Figure 3C). At the follow-up scan 9-months post-injury, significant activity was detected in both the sound and speech perception conditions (Figure 3B), with peaks in the bilateral auditory cortex and superior temporal gyrus (Table A2 in Appendix A). The strength of this activity was once again comparable to the neural responses observed in healthy controls in a prior study [13] (Figure 3C). There was no difference in activity for sound perception between the recovery and acute scan. In the speech perception condition, there was greater neural activity in the left superior temporal gyrus at recovery (Figure 4, Table A3 in Appendix A).

### 3.2. Command Following

Acutely, the patient showed no significant cortical activity in response to the spatial navigation task. However, when comparing activity between the recovery and acute scans, significant changes were observed in the left parahippocampal gyrus (Figure 5A, Table A3 in Appendix A). The strength of this activity was within the range of the activity observed in previously published healthy controls [20] at the single-subject level (Figure 5B).

### 3.3. Resting State Scan

Acutely, the mean correlation of the patient′s BOLD activity across all 10 resting state networks to the spatial template was *r* = 0.21 (SD = 0.11). At recovery, the mean correlation was *r* = 0.35 (SD = 0.17). We observed a significant increase in the correlation to the normal template averaged across the 10 resting state networks at recovery (*p* = 0.030), as seen in Figure 6A. Correlation to the template at recovery increased by at-least twofold from the acute scan in the visual medial, default mode, sensorimotor, and auditory network (Figure 6B).

## 4. Discussion

In this case report, we present a patient who sustained a severe TBI with DAI. The patient′s cerebral activity was evaluated using functional neuroimaging while they were critically ill and undergoing intensive care treatment. The patient completed follow-up imaging at functional recovery, 9-months post-injury. We report preserved auditory processing acutely and an increase in resting state connectivity and command-following abilities at functional recovery.

Acutely and at recovery, the patient had robust BOLD responses to the sound and speech perception contrasts. Peak activity was detected bilaterally in the primary auditory cortex and superior temporal gyrus—regions known to be involved in sound and speech perceptual processing [23,24,25]. The similarly robust magnitude and intensity of neural activity between the acute and recovery scan suggests that both sound and speech selective perceptual processing was unaltered while the patient was behaviourally unresponsive. Despite appearing clinically comatose, the location and degree of activity in the acute scan was comparable to that of healthy controls, suggesting that the neural machinery required to support sound and speech perception likely remained intact. This possibility is further supported by the minimal differences in activity observed between the recovery and acute scan. However, because the task was passive and no behavioural response was required, we cannot be sure that the neural responses elicited in the acute scan were indicative of the patient actively perceiving the auditory stimuli (that is, being aware of them). Moreover, the presence of such responses has been shown to be predictive of some level of recovery in both acute [13] and chronic [21] patients with disorders of consciousness. Given this preliminary study, we suggest that investigators should examine speech perception as the functional domain that may be most useful for prognostic purposes.

The presence of a robust BOLD response to the auditory perception task, accompanied by an inability to command follow acutely, suggests that this patient may have had preserved islands of residual cognitive function, as has been documented in chronic disorders of consciousness patients [26]. It has been proposed that many individuals with impairments in consciousness lack the necessary dynamic functional connections between cortical areas to sustain conscious behaviour [27]. This is supported by the low correlation value to the spatial template and the negative command following result while the patient was unresponsive and critically ill, which, taken together, suggests an absence of conscious awareness. Widespread cortical activity sustained through intact functional connections across multiple brain networks is thought, in part, to govern conscious behaviour [28,29]. Our data supports this theory as we observed an increase in the correlation between resting-state functional connectivity and the spatial template between the acute and the recovery scans, as well as increased activity in the command following task. Disrupted functional connections between cortical regions that then underwent partial restoration have been reported in a patient who emerged from a vegetative state to a minimally conscious state [30]. Our results suggest that functional reorganization also occurs during emergence from coma and that this reorganization allows for higher-order cognitive processing to occur, as seen in the positive command following result. In cases of *moderate* TBI, disruptions in connectivity have been tracked longitudinally from trauma onset and shown to persist for several months after injury [31,32]. Further longitudinal observations will be required to examine how the relationship between network abnormalities and neurobehavioral outcomes changes over the course of functional recovery. Using fMRI to characterize network changes over the course of functional recovery will contribute to our understanding of how brain activity changes at rest from a behaviourally unresponsive to a conscious state. Specifically, examining which networks undergo functional reorganization, and to what extent, will improve our knowledge about resting state recovery mechanisms.

Our study adds to the growing body of literature demonstrating the clinical utility of functional neuroimaging in the assessment of residual cerebral activity in critically ill patients. FMRI is emerging as a potential aid to current diagnostic and prognostic tools as it provides valuable insight into cerebral activity that may not be accessible through subjective bedside assessments of cortical function. With diffuse axonal injuries being the leading cause of posttraumatic comas and prolonged vegetative states [33], having an objective measure of residual cortical activity may assist clinical decision making. Functional neuroimaging research in acute critically ill brain injured patients is showing promise, with some studies predicting reversible impairments of consciousness from neural responses to passive stimuli [7,13]. Specifically, one study showed that patients with greater activity in the primary somatosensory cortex were more likely to recover consciousness, in comparison to those who succumbed to their injury [7]. While the stimuli used differs from ours, both studies show that fMRI can be used to assess sensory perception in behaviourally unresponsive patients. Another study used functional neuroimaging to assess sensory perception in an unresponsive severe TBI patient [34]. The fMRI scan showed preserved auditory perception and the patient subsequently regained conscious awareness. These reported findings are consistent with the results of our case study. Additionally, a recent study that used fMRI to detect consciousness in acute TBI patients showed no fMRI responses in the two comatose patients imaged [10]. Notably, the fMRI findings from our study also show no evidence of sustained conscious awareness. Additionally, the integrity of resting state networks in comatose patients has emerged as a potential biomarker for meaningful recovery as positive long-term outcomes have been associated with preservation of these networks following injury [8,9,12]. Thus, incorporating functional neuroimaging into the standard of care for critically ill brain injured patients is feasible and provides objective measures of cortical activity that may otherwise be difficult to obtain at the bedside. While fMRI may be a costly alternative, traditional clinical assessments may be confounded by the behavioural fluctuations of a patient and the low inter-rater reliability of results [4,5]. Additionally, severe traumatic injuries may prevent EEG leads from being placed on a patient’s head. FMRI overcomes these issues as it can provide objective measures of a patient′s neural activity in response to stimuli and at rest. To mitigate the high costs of fMRI, we suggest, where at all feasible, that functional MRI scans be combined with clinically indicated structural MRI scans. This also mitigates the risks of multiple visits to the MRI.

Nevertheless, while the research is promising, before these techniques can be widely adopted clinically, many technical and ethical hurdles will still need to be overcome [35,36]. For example, the presence of intubation, parenteral nutrition supports, and venous lines in critically ill patients creates an added complexity that occurs less frequently in chronic disorders of consciousness cases. During fMRI imaging, patients should be closely monitored by the treating medical team and accompanied by a nurse and respiratory technologist. If extensive motion is observed, cyclical signals such as respiratory and intra-venous pump rates may be regressed out during the data analyses. Non-essential devices should not accompany the patient to further reduce motion, such as nutritional feeds, which can be put on hold until imaging is performed. Additionally, the use of block designs with repeated trials may help reduce artifacts resulting from medical equipment [37]. Furthermore, sedation may be required in patients with uncontrolled and sporadic movements while in the scanner, which would make it difficult to reliably analyze and interpret data. Promisingly, the effects of sedation on resting state MRI are relatively minimal for some cortical networks in disorders of consciousness patients who have sustained a traumatic brain injury [38].

While this study is one of the first to assess cerebral function in a severe TBI patient acutely after injury and at functional recovery, there are notable limitations. It was not possible to scan the patient earlier than 26 days post-injury because of his raised intracranial pressure and inability to lie flat in the scanner prior to that point. From initial admission to the date of functional neuroimaging, the patient′s GCS improved from 4T E(1), V(1T), M(2), to 8T E(4), V(1T), M(3). Although a patient with a GCS of 8 or less is clinically considered to be in a coma [39], the patients spontaneous eye opening on the same day of the first fMRI scan suggests that he may have been progressing into a state of unresponsive wakefulness syndrome. Additionally, the patient’s level of consciousness was assessed with the GCS at the bedside, which lacks robust sensitivity. Future studies should use more sensitive scales such as the Coma Recovery Scale-revised [40]. Furthermore, the results presented in this case study should be interpreted with caution as they inevitably lack generalizability to a wider population. Ultimately, a large patient population for a within-subject longitudinal study is needed to elucidate how cerebral activity changes from a behaviourally unresponsive state to meaningful recovery in survivors of severe TBI.

## 5. Conclusions

In this work, we described a patient who sustained a severe TBI resulting in coma. Cerebral activity was assessed using functional neuroimaging while the patient was behaviourally unresponsive and critically ill and at 9-months post-injury, when a clinically meaningful recovery was made. We observed preserved sound and speech processing in the acute unresponsive state and an increase in resting state connectivity and command-following abilities at functional recovery. These results add to the growing body of literature that demonstrates the feasibility of assessing cerebral activity in acute coma using functional neuroimaging and describes how neural activity changes from acute injury to meaningful recovery in a single patient.

## Figures and Tables

**Figure 1 brainsci-10-00604-f001:**
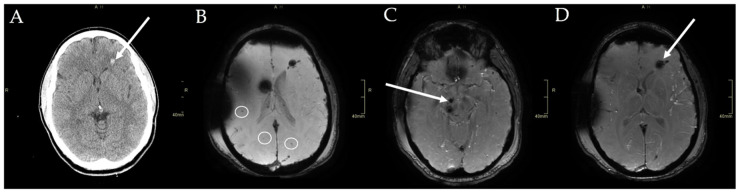
Structural imaging of the reported patient. (**A**) CT imaging showed preserved grey and white matter differentiation with no major mass lesion or mass effect. A small left subcortical frontal hemorrhage was seen and associated with diffuse axonal injury. (**B**) Susceptibility weighted imaging showed punctate foci of parenchymal susceptibility (indicated by the circles) in both cerebral hemispheres, consistent with diffuse axonal injury *. (**C**) Susceptibility weighted imaging revealed a unilateral right sided midbrain injury. There were punctate foci of parenchymal susceptibility from blood in the right cerebral peduncle, as indicated by the white arrow *. (**D**) Susceptibility weighted imaging showed punctate foci of parenchymal susceptibility from blood in the left subcortical frontal lobe, as indicated by the white arrow *. * Black shaded areas are a result of motion artifact.

**Figure 2 brainsci-10-00604-f002:**
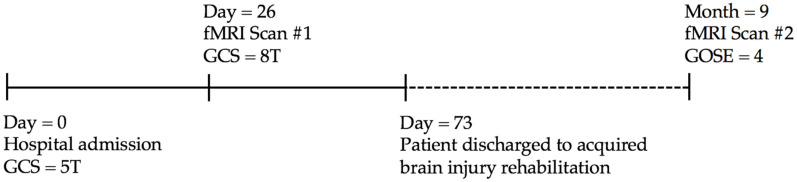
Schematic timeline of the patient from acute care admission to the follow-up fMRI scan. Written, informed consent was obtained from the patient′s substitute decision-maker for the first fMRI scan and subsequently, from the patient once consciousness and capacity were regained.

**Figure 3 brainsci-10-00604-f003:**
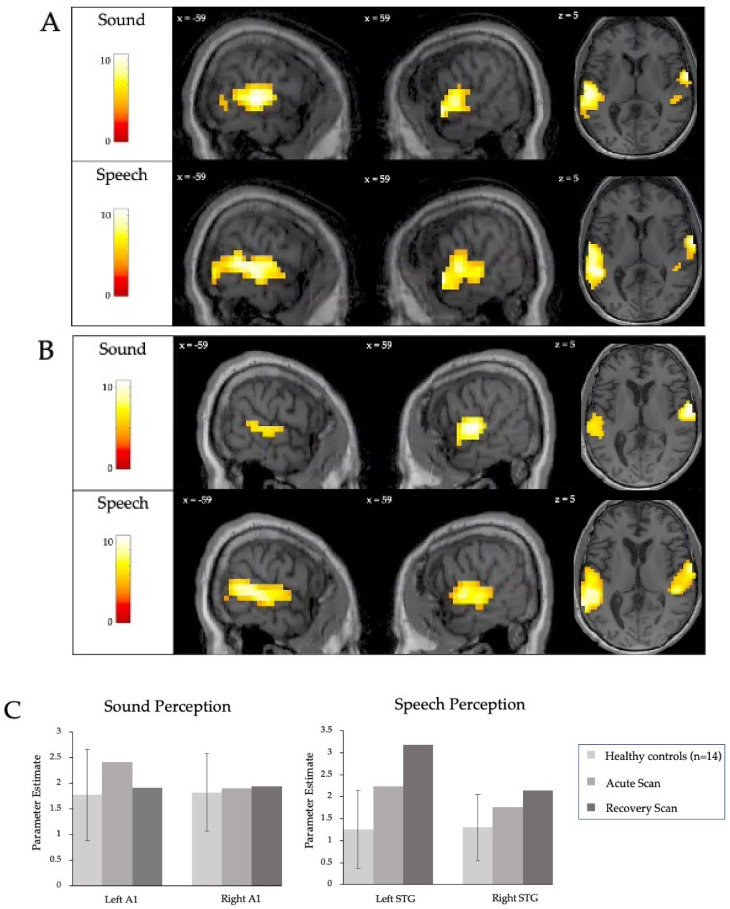
Auditory perception acutely and at recovery. (**A**) Neural responses to the auditory perception paradigm in acute brain injury *. (**B**) Neural responses to the auditory perception paradigm 9-months post-injury *. (**C**) The patterns of activity in the auditory perception paradigm corresponding to the peak voxel co-ordinates in each contrast. The acute scan and recovery scan parameters are plotted adjacent to the mean activity of 14 healthy participants [13]. The error bars indicate the standard deviation of the healthy controls. * Individual patient results are thresholded at *p* < 0.05, FWE corrected for multiple comparisons.

**Figure 4 brainsci-10-00604-f004:**
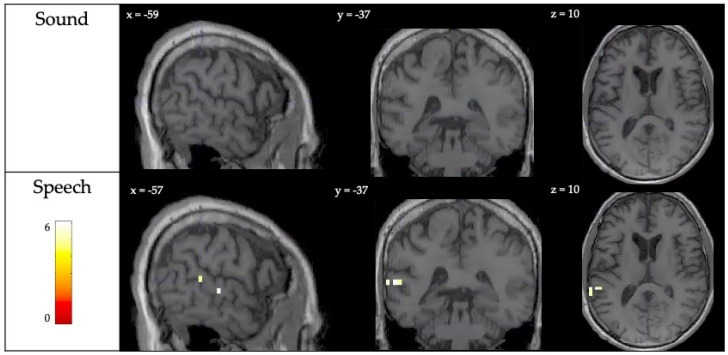
Neural responses to the auditory perception paradigm when comparing differences in activity between the recovery and acute scan (Recovery > Acute). Individual patient results are thresholded at *p* < 0.05, FWE corrected for multiple comparisons.

**Figure 5 brainsci-10-00604-f005:**
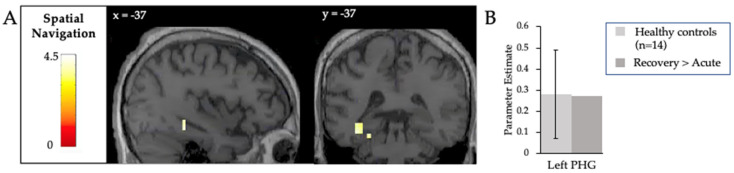
(**A**) Neural responses to the spatial navigation paradigm when comparing the activity of the recovery to the acute scan. Individual patient results are thresholded at *p* < 0.001. Images are masked inclusively by the group analysis of healthy control participants. (**B**) The difference in activity in the left parahippocampal gyrus is plotted adjacent to the mean activity observed in this region in 14 healthy participants performing the same task [20]. The error bars indicate the standard deviation of the healthy controls.

**Figure 6 brainsci-10-00604-f006:**
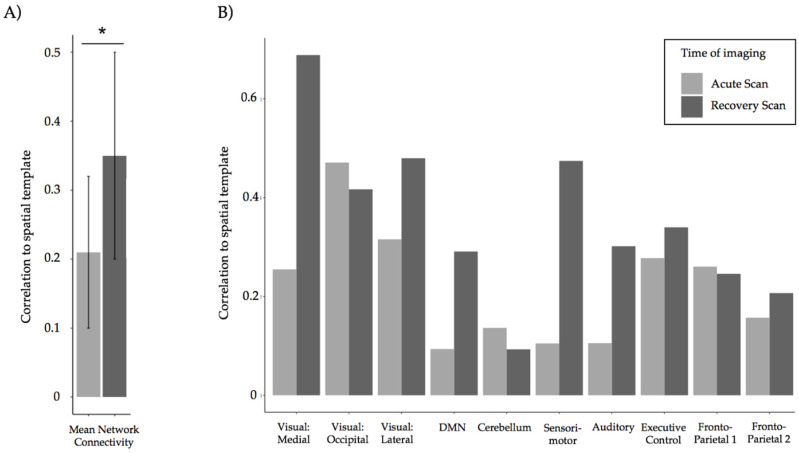
Resting state connectivity acutely and at recovery. (**A**) A significant increase in correlation to the spatial template averaged across 10 resting state networks were observed at recovery. The * denotes a significant difference in correlation between the two time points. (**B**) Increases by at least two-fold in connectivity were observed in the visual medial, default mode, sensorimotor, and auditory networks at recovery.

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
