# Peer review of "Cortical Function in Acute Severe Traumatic Brain Injury and at Recovery: A Longitudinal fMRI Case Study"

_brainsci, 2020, doi:10.3390/brainsci10090604_

Round 1

Reviewer 1 Report

Kazazian and colleagues describe a clinical case of a comatose patient which was evaluated with fMRI and an auditory/speech paradigm at two different windows of time during his recovery. Their main findings are a relatively well-preserved BOLD response to sound and speech in both scans, followed by an increase in the mean network connectivity between the two scans, which is concordant with signs of clinical recovery.

The manuscript is well written, the clinical characterization and research methods are explained correctly. The results are straightforward and the conclusions are sounded. The authors also properly discuss the limitations of their study. As such, I think it is a very nice manuscript.

I suggest the authors expand a bit the discussion section, to make it more scholar, incorporating other similar studies and analyzing their similitudes and differences with their own study. 

Author Response

Reviewer #1: Comments to the Author

1) I suggest the authors expand a bit in the discussion section, to make it more scholarly, incorporating other similar studies and analyzing their similitudes and differences with their own study.

Our response:

Thank you for your suggestion to incorporate other similar studies into the discussion. We have now expanded this section (see lines 301-310) to compare our findings more explicitly with other studies that have used fMRI in acute comatose patients. We believe the added points have strengthened our discussion and helped us to improve the clarity and impact of our manuscript and hope it is now ready for publication.  

Below is the statements added on lines 301-310:

Specifically, one study showed that patients with greater activity in the primary somatosensory cortex were more likely to recover consciousness, in comparison to those who succumbed to their injury [7]. While the stimuli used in that study differs from ours, both show that fMRI can be used to assess sensory perception in behaviourally unresponsive patients. Another study used functional neuroimaging to assess sensory perception in an unresponsive severe TBI patient [34]. The fMRI scan showed preserved auditory perception, and the patient subsequently regained conscious awareness. These findings are consistent with the results of our case study. Additionally, a recent study that used fMRI to detect covert consciousness in acute TBI patients showed negative fMRI responses in the two comatose patients imaged [10]. Notably, the fMRI findings from our study also show no evidence of sustained conscious awareness in acute coma.

Reviewer 2 Report

This is an interesting case report describing data from a longitudinal analysis (2 time points) of a severe TBI case underwent to fMRI session to detect possible indications or measurements about the residual brain functioning post-TBI during the "acute" phase vs. "chronic" phase. The manuscript is well written and the experimental design is proper. However, these are the main concerns for this reviewer:

1) The first fMRI assessment was done after 26 days post-TBI, so it would be hard to consider this time-period as the "acute" moment of the TBI. Maybe the authors could avoid to sue the wards acute vs. chronic and just describe an earlier vs and longer time point of fMRI assessment. 

2) Being a car accident, did the authors excluded the use of alcohol or drug abuse at the moment of the accident (the car went on a tree, so the subject maybe was in an altered status of consciousness before the actual TBI).

3) Of course, these types of assessment (fMRI) imply a lot of technical difficulties and possible artifacts due to the presence of intubation, parenteral nutrition supports, venous lines, etc. Could the authors propose how to solve these technical issues in a broader clinical context?

4) In economic terms, which is the cost/benefit ratio between the us of fMRI data vs. clinical assessment+neurophysiology methods, which are much way cheaper than running an fMRI session?

5) Could the author suggest which one among the functional domains that they assessed if there would be one more useful in terms of possible prognosis aspects?

6) How DAI was identified on MRI? Figure1B does not clearly show that.

7) Could the authors speculate on how fMRI data could help to improve our knowledge about resting-state recovery mechanisms? Just one statement in the discussion could be sufficient.

Round 2

Reviewer 2 Report

The authors replies were mostly satisfying. Although some concerns remain about the "acute period" definition (this reviewer recognize though, that the authors made numerous changes to clarify this point across the entire manuscript) and the punctate foci of parenchyma as lesions consistent with DAI (this correlation has never been confirmed by specific neuropathological assessments), this reviewer thinks and recommends that the manuscript should be accepted for publication. 

Author Response

Thank you for recommending the manuscript should be accepted for publication.